# Magnitude of asymptomatic COVID-19 cases throughout the course of infection: A systematic review and meta-analysis

Muluneh Alene[1]*, Leltework Yismaw[1], Moges Agazhe Assemie[1], Daniel Bekele Ketema[1], Belayneh Mengist[1], Bekalu Kassie[2], Tilahun Yemanu Birhan[3]

1 Department of Public Health, Debre Markos University, Debre Markos, Ethiopia, 2 Department of Midwifery, Debre Markos University, Debre Markos, Ethiopia, 3 Department of Epidemiology and Biostatistics, University of Gondar, Gondar, Ethiopia

* mulunehadis@gmail.com

## Abstract

### Background

Asymptomatic SARS-CoV-2 infections are responsible for potentially significant transmission of COVID-19. Worldwide, a number of studies were conducted to estimate the magnitude of asymptomatic COVID-19 cases. However, there is a need for more robust and well-designed studies to have a relevant public health intervention. Synthesis of the available studies significantly strengthens the quality of evidences for public health practice. Thus, this systematic review and meta-analysis aimed to determine the overall magnitude of asymptomatic COVID-19 cases throughout the course of infection using available evidences.

### Methods

We followed the PRISMA checklist to present this study. Two experienced review authors (MA and DBK) were systematically searched international electronic databases for studies. We performed meta-analysis using R statistical software. The overall weighted proportion of asymptomatic COVID-19 cases throughout the course infection was computed. The pooled estimates with 95% confidence intervals were presented using forest plot. Egger's tests were used to assess publication bias, and primary estimates were pooled using a random effects model. Furthermore, a sensitivity analysis was conducted to assure the robustness of the result.

### Results

A total of 28 studies that satisfied the eligibility criteria were included in this systematic review and meta-analysis. Consequently, in the meta-analysis, a total of 6,071 COVID-19 cases were included. The proportion of asymptomatic infections among the included studies ranged from 1.4% to 78.3%. The findings of this meta-analysis showed that the weighted pooled proportion of asymptomatic COVID-19 cases throughout the course of infection was 25% (95%CI: 16–38). The leave-one out result also revealed that the weighted pooled average of asymptomatic SARS-CoV-2 infection was between 28% and 31.4%.

**Data Availability Statement:** All relevant data are within the manuscript and its Supporting Information files.

**Funding:** The authors received no specific funding for this work.

**Competing interests:** The authors have declared that no competing interests exist.

**Abbreviations:** COVID-19, The 2019 Novel Coronavirus Disease; MERS-CoV, Middle East respiratory syndrome coronavirus; SARS-CoV, Severe Acute Respiratory Syndrome Coronavirus; Sever SARS-CoV-2, Severe Acute Respiratory Syndrome Coronavirus 2; WHO, World Health Organization; CI, Confidence Interval.

## Conclusions

In conclusion, one-fourth of SARS-CoV-2 infections are remained asymptomatic throughout the course infection. Scale-up of testing, which targeting high risk populations is recommended to tackle the pandemic.

## Background

The novel coronavirus disease 2019 ("COVID-19") caused by sever acute respiratory syndrome coronavirus 2 (SARS-CoV-2) has now established a global pandemic [1]. The pandemic is a challenge for both developed and developing countries causing huge stress on the healthcare system of all countries [2]. Up to February 20, 2021; there were more-than 110 million total COVID-19 cases with more-than two million deaths, worldwide [3]. As studies reported, the early sign of the COVID-19 is pneumonia [4]. In addition, of COVID-19 patients who developed signs and symptoms, the most frequently reported symptoms was fever followed by cough [5].

Asymptomatic and presymptomatic COVID-19 cases are responsible for potentially significant transmissions, and this makes a challenge to control the pandemic [6]. Approximately, half of individuals with positive test results don't have any symptoms at the time of testing [7]. Additionally, about one-fifth of SARS-CoV-2 infections are remained asymptomatic throughout the course infection. One study indicated that the viral shedding coronavirus is peaked on before symptom onset [8]. This indicates that a substantial proportion of transmission probably occurred before first symptoms in the primary case. Another studies conducted in Singapore and China revealed that about 6.4%, and 12.6% of COVID-19 cases were attributed to asymptomatic transmission, respectively. Consequently, asymptomatic transmission expected to occur 1–3 days before symptom onset of source patients [9].

The serial interval and incubation period are the two main epidemiological parameters that suggests for presymptomatic transmission of COVID-19. When the average serial interval of COVID-19 is shorter than the average incubation period, some proportion of cases are attributed to presymptomatic transmission [10]. Accordingly, an observational study which aimed to provide the epidemiological parameters of COVID-19 using seven countries data revealed that the mean incubation period and serial interval were 7.44 days and 6.70 days, respectively [11].

Globally, a number of studies were conducted to determine the magnitude of asymptomatic SARS-CoV-2 infection. However, there is a need for more robust and well-designed studies to have relevant public health intervention. Studies vary depending on the number of participants recruited, the type of design employed and the country in which the study conducted [12, 13]. Combined findings of existing studies significantly strengthen the quality of evidences for public health practice. Since the biological, clinical and epidemiological characteristics of COVID-19 didn't well known, this well design and appropriately performed study is needed to have a solid evidence-based intervention. The findings of this study will have a potential role to inform policymakers and stakeholders to combat the pandemic. Thus, this systematic review and meta-analysis aims to determine the pooled magnitude of asymptomatic COVID-19 cases throughout the course of infection using existing evidences.

## Methods

### Searching for studies

The study follows the preferred reporting items for systematic review and meta-analysis (PRISMA) to present this study. The search was done by two experienced review authors (MA

and MAA) from international electronic databases (Google Scholar, PubMed, Science Direct, Web of Science, and CINAHL). In addition, we searched from the reference lists of the included studies to identify any other studies that may have been missed by our search strategy. We used the following search terms: "magnitude" OR "prevalence" AND "asymptomatic" OR "presymptomatic" OR "silent" AND "transmission" AND "coronavirus OR "COVID-19" OR "novel coronavirus" OR "SARS-CoV-2". Our search was performed between the 1st of June and the 9th of December, 2020. Finally, all studies were imported into reference management software (Mendeley desktop).

## Inclusion criteria

**Estimates reported**: all observational studies reported the magnitude of asymptomatic COVID-19 cases throughout the course of infection

   **Study setting**: worldwide

   **Population**: all age group

   **Publication status**: all published, and unpublished articles

   **Language**: only studies reporting using English language

   **Publication date**: published from the 1st of January to the 9th of December, 2020

## Exclusion criteria

Articles that was not report the outcome of interest, case reports and review studies were excluded.

## Outcome variable and data extraction

The outcome variable of this study was the magnitude of true asymptomatic SARS-CoV-2 infection. Asymptomatic SARS-CoV-2 infection is defined as an individual without a history of clinical signs and symptoms throughout the course of infection. Two experienced review authors (MA and LY) extracted all essential data from the included studies using a predesigned data extraction form. The data extraction form organized as; the last name of the first author, the country of the study conducted, data collection period, sample size, magnitude of asymptomatic COVID-19 cases. Any inconsistencies in the data extraction process were decided through discussion involving all authors.

## Quality assessment

Two review authors (LY and TYB) were assessed the risk of bias of the included articles. The Newcastle Ottawa Scale (NOS) adapted for cross-sectional studies was used to evaluate the quality of studies [14]. This tool organized from three major sections. Consequently, the first section scored on the basis of one to five stars focuses on the methodological quality of each study. The second segment of the tool evaluates the comparability of the study groups with a maximum possibility of two stars to be given. The last section of the tool is concerned with the outcomes and statistical analysis of the included studies with a maximum possibility of three stars to be given. Each author rated the quality of each article. Any inconsistent report between the two reviewers was decided by taking the average score of the two reviewers'. Finally, the assessed articles with a score of less than six out of ten were considered as achieving low quality.

## Data processing and analysis

After extracting all essential data using Microsoft Excel, data were exported to R statistical software for further analysis. In-consistency among the reported magnitude of asymptomatic

SARS-CoV-2 infection was assessed using $I^2$-index [15]. To estimate the weighted pooled magnitude of asymptomatic COVID-19 cases, a random-effect meta-analysis with an estimation of DerSimonian and Laird method was performed. The publication bias was assessed using a tool known as a funnel plot. Funnel plot asymmetry was also tested by using Egger's and Beggs' tests [16]. Furthermore, leave-one-out meta-analysis was conducted to assure the robustness of the result. Leave-one-out analysis involves performing a meta-analysis on each subset of the studies obtained by leaving out exactly one study. This shows how each individual study affects the overall estimate of the rest of the studies.

## Results

### Search results

**Fig 1,** shows the flow chart diagram describing the selection of studies included in the systematic review and meta-analysis. Our search resulted with a total of 8,260 studies. Consequently, 134 articles were eligible for screening after excluding duplication. Ninety six articles were excluded after reading the title and abstract. After carefully assessed the text in the included studies, ten articles were removed due to not extractable result and the outcome of interest. Finally, in this study, we included a total of 28 studies that satisfied the eligibility criteria.

### Description of the included studies

The detail description of the included studies are presented in (**Table 1**). In this systematic review and meta-analysis, a total of 28 studies with a total COVID-19 cases of 6,071 COVID-19 included. The smallest sample size was 23 [17], while the largest sample size was 712 [18]. Nearly half (48.3%) of the included studies were conducted from China.

### Magnitude of asymptomatic COVID-19 cases

Of the included studies, the proportion of asymptomatic SARS-CoV-2 infections ranged from 1% to 81%. Consequently, our meta-analysis showed that the weighted pooled truly asymptomatic COVID-19 cases was 25% (95%CI: 16–38) (**Fig 2**). **Table 2,** shows the sensitivity analysis of the study. The minimum weighted pooled proportion (28%) of SARS-CoV-2 was found by removing article [22], while the maximum proportion (31.4%) was obtained after removing the study [28]. The issue of publication bias was assessed by graphic inspection of funnel plot and using the rank correlation test. Even though, the funnel plot looks asymmetrical (**Fig 3**), the rank correlation test showed that no relationship between the effect size and its precision (P-value = 0.4).

## Discussion

COVID-19 pandemic remains a major public health problem worldwide. Currently, there is no enough evidences to recommend any specific medication for the treatment of COVID-19. Presymptomatic and asymptomatic SARS-CoV-2 infections are capable to transmitting the virus, and this makes challenging to prevent and control the pandemic. Previous evidences showed that SARS-CoV-2 infections are spread more rapidly compared with Sever Acute Respiratory Syndrome Coronavirus (SARS-CoV) and Middle East Respiratory Syndrome Coronavirus (MERS-CoV) [45]. Previous studies suggest that contact and symptom based screening might fail to identify all potential SARS-CoV-2 infections. The current study was aimed to determine the overall magnitude of asymptomatic COVID-19 cases throughout the course of infection.

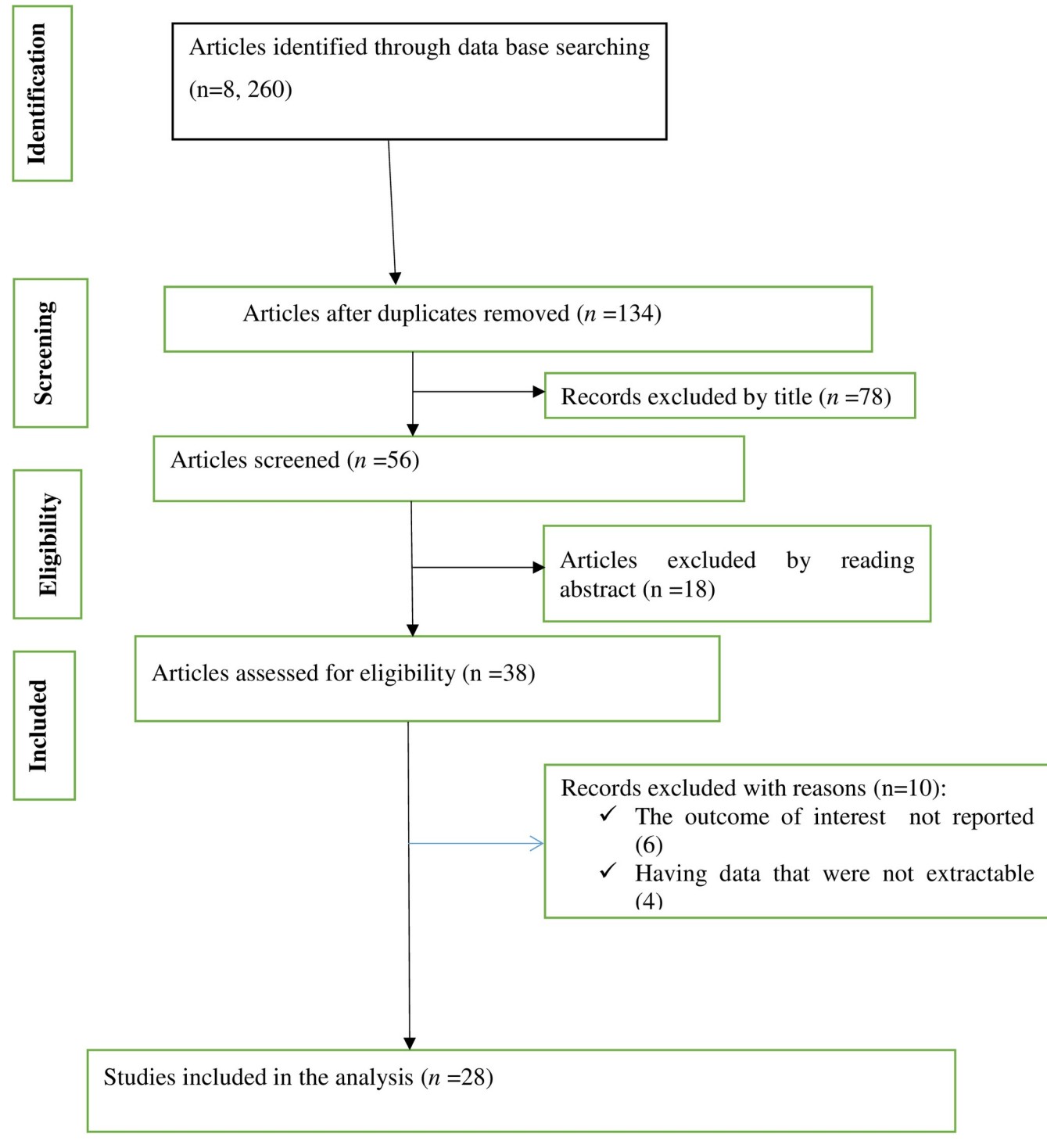

**Fig 1. Flow chart diagram describing the selection of studies included in the systematic review and meta-analysis on the magnitude of truly asymptomatic SARS-CoV-2 infection, 2020.**

We computed the overall weighted proportion of COVID-19 cases who hadn't developed signs and symptoms throughout the course of infection. Accordingly, one-fourth (25% (95% CI: 16–38)) of COVID-19 cases were asymptomatic throughout the course of infection. This

**Table 1. Descriptions of the included studies conducted on the proportion of asymptomatic SARS-CoV-2 infection, 2020.**

| No. | First author | Country | Study period | Total cases | Asymptomatic cases |
|---|---|---|---|---|---|
| 1. | An et al [19] | China | April, 2020 * | 25 | 16 |
| 2. | Arons et al [20] | USA | March 13 to 20,2020 | 48 | 3 |
| 3. | Chun et al [21] | South Korea | January 23 to March 31, 2020 | 89 | 16 |
| 4. | Day et al [22] | China | April 1,2020 * | 166 | 130 |
| 5. | Feaster et al [23] | USA | April, 2020 * | 631 | 257 |
| 6. | Inui et al [24] | Japan | February 7 to 28,2020 | 104 | 76 |
| 7. | Keeley et al [25] | Argentina | March, 2020 * | 128 | 104 |
| 8. | Kimball et al [17] | USA | March 13,2020 * | 23 | 3 |
| 9. | Kong et al [26] | China | January 25 to February 20,2020 | 511 | 100 |
| 10. | Lavezzo et al [27] | Italy | February 23 to March 8, 2020 | 102 | 44 |
| 11. | Ling et al [28] | China | January 23 to February 18, 2020 | 295 | 4 |
| 12. | Long et al [29] | China | February 6, 2020 * | 178 | 37 |
| 13. | Luo et al [30] | China | February 21, 2020 * | 83 | 8 |
| 14. | Ma et al [31] | China | Jan 23 to March 10,2020 | 47 | 11 |
| 15. | Meng et al [32] | China | Jan 1 and Feb 23, 2020 | 58 | 42 |
| 16. | Mizumoto et al [33] | Japan | February 20,2020 * | 634 | 328 |
| 17. | Moriarty et al [18] | Japan | February to March 2020 | 712 | 331 |
| 18. | Nishiura et al [34] | Japan | February 12, 2020 * | 565 | 235 |
| 19. | Noh et al [35] | North Korea | March, 2020 * | 199 | 53 |
| 20. | Rivett et al [36] | United kingdom | April, 2020 * | 30 | 17 |
| 21. | Tabata et al [37] | Japan | Feb 11 to Feb 25, 2020 | 104 | 33 |
| 22. | Tian et al [38] | China | Feb 10, 2020 * | 262 | 13 |
| 23. | Wan et al [39] | China | February 20,2020 * | 78 | 2 |
| 24. | Wong et al [40] | Brunei | April 24, 2020 * | 138 | 16 |
| 25. | Xu et al [41] | China | January 18 to February 26, 2020 | 342 | 15 |
| 26. | Zhao et al [42] | China | February 21,2020 * | 160 | 4 |
| 27. | Zhou et al [43] | China | March 4, 2020 * | 328 | 10 |
| 28. | Zhou et al [44] | China | Jan 23 to March 3,2020 | 31 | 9 |

*study period was not clearly stated.

result is comparable with previous study conducted on the asymptomatic SARS-CoV-infections. However, this result is lower than previous studies conducted on asymptomatic SARS-CoV-2 infection [6, 7]. The possible reason for this variation might be asymptomatic COVID-19 cases considered in the previous studies will develop sign and symptoms during hospitalization period. A study conducted in Barcelona, Spain revealed that more-than two-third of SRAS-CoV-2 infections are asymptomatic at the time of testing [46].

We also compared the proportion of asymptomatic SARS-CoV-2 infections with previously emerged coronavirus outbreaks. A study that reports the role of asymptomatic patients in the transmission of MERS-CoV showed that one-fourth (25.1%) of MERS-CoV were asymptomatic [47]. Though, one study showed that small number of SARS-CoV cases are asymptomatic, unlike COVID-19, sever acute respiratory syndrome could be controlled by effective isolation of symptomatic patients [48]. One study also showed that nearly half of SARS-CoV patients were asymptomatic throughout the course hospitalization [49]. This inconsistency might be due to the difference in clinical severity between COVID-19 cases, MERS and SARS-CoV [50].

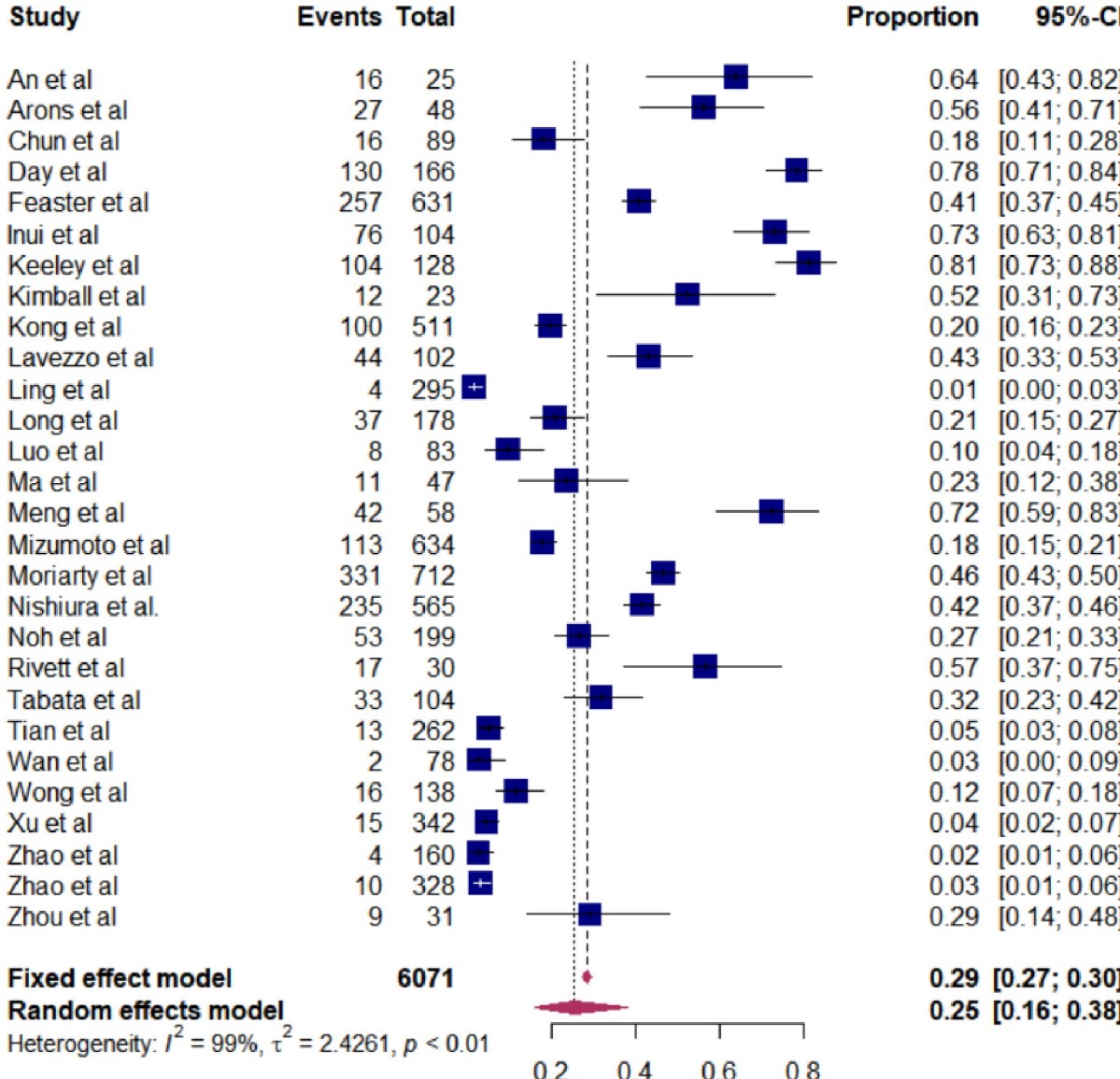

**Fig 2. Forest plot that shows the weighted pooled proportion of asymptomatic SARS-CoV-2 infection using available studies, 2020.**

## Limitation

The current study has a number of limitations. Firstly, the majority of the included studies had relatively small sample size which may decrease the power of the study. Secondly, the review was limited to only articles published in the English language. Lastly, since the included articles are limited to few countries, it may not represent the global figure of asymptomatic SARS-CoV-2 infection.

## Conclusions

In conclusion, one-fourth of SARS-CoV-2 infections are remained asymptomatic throughout the course infection. Scale-up of testing, which targeting high risk populations is recommended to tackle the pandemic.

**Table 2. The sensitivity analysis to estimate the pooled proportion of truly asymptomatic SARS-CoV-2 infection, 2020.**

| No. | Study omitted | Pooled proportion (95%CI) | No. | Study omitted | Pooled proportion (95%CI) |
|---|---|---|---|---|---|
| 1. | An et al | 28.7 (18.9, 39.6) | 15. | Meng et al | 28.4 (18.8,39.0) |
| 2. | Arons et al | 28.9 (19.0, 39.9) | 16. | Mizumoto et al | 30.3 (20.1,41.6) |
| 3. | Chun et al | 30.3 (20.1,41.5) | 17. | Moriarty et al | 29.2 (19.2,40.4) |
| 4. | Day et al | 28 (18.6,38.5) | 18. | Nishiura et al | 29.4 (19.3,40.6) |
| 5. | Feaster et al | 29.4 (19.3,40.6) | 19. | Noh et al | 30.0 (19.8,41.2) |
| 6. | Inui et al | 28.3 (18.7,38.9) | 20 | Rivett et al | 29.0 (19.1,39.9) |
| 7. | Keeley et al | 27.9 (18.6,38.3) | 21. | Tabata et al | 29.8 (19.6,41.0) |
| 8. | Kimball et al | 29.1 (19.2,40.2) | 22. | Tian et al | 31.1 (20.9,42.1) |
| 9. | Kong et al | 30.3 (20.1,41.5) | 23. | Wan et al | 31.2 (21.2,42.2) |
| 10. | Lavezzo et al | 29.4 (19.3,40.5) | 24. | Wong et al | 30.6 (20.5,41.8) |
| 11. | Ling et al | 31.4 (21.5,42.5) | 25. | Xu et al | 31.1 (21.1,42.1) |
| 12. | Long et al | 30.2 (20.0,41.5) | 26. | Zhao et al | 31.3 (21.3,42.2) |
| 13. | Luo et al | 30.7 (20.6,41.9) | 27. | Zhou et al | 31.2 (21.2,42.2) |
| 14. | Ma et al | 30.1(19.9,41.3) | 28. | Zhou et al | 29.8 (19.7,41.1) |

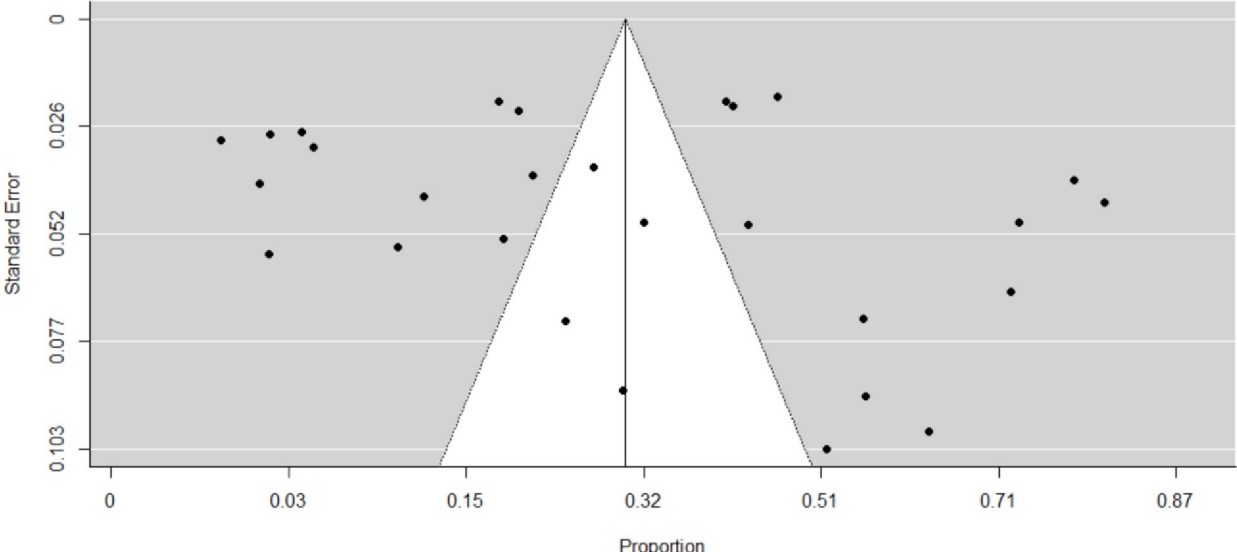

**Fig 3. Funnel plot to check the publication bias of the included studies conducted on truly asymptomatic SARS-CoV-2 infection, 2020.**

## Supporting information

**S1 Checklist. PRISMA 2009 checklist.**
(DOC)

**S1 Table. Individual effect sizes of the included studies conducted on truly asymptomatic SARS-CoV-2 infection, 2020.**
(DOCX)

## Author Contributions

**Conceptualization:** Muluneh Alene, Daniel Bekele Ketema, Belayneh Mengist.

**Data curation:** Muluneh Alene, Leltework Yismaw, Moges Agazhe Assemie, Belayneh Mengist, Bekalu Kassie, Tilahun Yemanu Birhan.

**Formal analysis:** Muluneh Alene, Leltework Yismaw, Moges Agazhe Assemie, Daniel Bekele Ketema, Belayneh Mengist, Bekalu Kassie, Tilahun Yemanu Birhan.

**Investigation:** Leltework Yismaw, Moges Agazhe Assemie, Daniel Bekele Ketema, Bekalu Kassie, Tilahun Yemanu Birhan.

**Methodology:** Muluneh Alene, Leltework Yismaw, Moges Agazhe Assemie, Daniel Bekele Ketema, Belayneh Mengist, Tilahun Yemanu Birhan.

**Software:** Muluneh Alene.

**Supervision:** Moges Agazhe Assemie, Daniel Bekele Ketema, Bekalu Kassie, Tilahun Yemanu Birhan.

**Validation:** Muluneh Alene, Leltework Yismaw, Daniel Bekele Ketema, Belayneh Mengist, Bekalu Kassie, Tilahun Yemanu Birhan.

**Visualization:** Tilahun Yemanu Birhan.

**Writing – original draft:** Muluneh Alene, Moges Agazhe Assemie, Daniel Bekele Ketema, Belayneh Mengist.

**Writing – review & editing:** Muluneh Alene, Leltework Yismaw, Moges Agazhe Assemie, Belayneh Mengist, Bekalu Kassie, Tilahun Yemanu Birhan.

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
