## [Decision Letter · Decision Letter 0]

30 Oct 2020

PONE-D-20-28251

Magnitude of asymptomatic COVID-19 cases throughout the course of infection: a systematic review and meta-analysis

PLOS ONE

Dear Dr. Muluneh Alene Addis,

Thank you for submitting your manuscript to PLOS ONE. After careful consideration, we feel that it has merit but does not fully meet PLOS ONE’s publication criteria as it currently stands. Therefore, we invite you to submit a revised version of the manuscript that addresses the points raised during the review process.

We look forward to receiving your revised manuscript.

Kind regards,

Kin On Kwok, Ph.D

Academic Editor

PLOS ONE

Journal Requirements:

2. Please confirm that you have included all items recommended in the PRISMA checklist including the full electronic search strategy used to identify studies with all search terms and limits for at least one database. Please attach a Supplemental file of the results of the individual components of the quality assessment, not just the overall score, for each study included. See https://journals.plos.org/plosmedicine/article?id=10.1371/journal.pmed.1000100#pmed-1000100-t003 for guidance. Thank you.

3. Please ensure that you refer to Figure 2 in your text as, if accepted, production will need this reference to link the reader to the figure.

Additional Editor Comments (if provided):

This article aims to assess the magnitude of asymptomatic and pre-symptomatic COVID-19 cases which is essential and currently relevant for COVID-19 pandemic. Search terms may need to be refined to include as many as articles in the initial search. Will the authors consider to have pooled estimate stratified by age, setting such as hospital or community and study period to enrich the content. Also the exclusion criteria was not explicit mentioned. In terms of the language, authors are suggested to have substantial edit to arrive at the publishable quality.

Reviewers' comments:

Reviewer's Responses to Questions

**Comments to the Author**

1. Is the manuscript technically sound, and do the data support the conclusions?

Reviewer #1: Yes

Reviewer #2: Partly

2. Has the statistical analysis been performed appropriately and rigorously? 

Reviewer #1: Yes

Reviewer #2: Yes

3. Have the authors made all data underlying the findings in their manuscript fully available?

Reviewer #1: Yes

Reviewer #2: Yes

4. Is the manuscript presented in an intelligible fashion and written in standard English?

Reviewer #1: Yes

Reviewer #2: No

5. Review Comments to the Author

Reviewer #1: Background

- Severe Acute Respiratory Syndrome Corn babirusa 2, instead of Sever respiratory syndrome Coronavirus 2

- This systematic review and meta-analysis aims to determine the pooled magnitude of asymptomatic COVID-19 cases throughout the course of infection? What about total cases, pre-symptomatic cases? The course of infection or the course of outbreak?

Method

- Additional search terms: cases, incidence, proportion, incubation period, serial interval

- Systematic review does not only include observational studies

- Study setting: worldwide? At hospital? At community?

- Population: All age group? Please provide age range.

- Publication status: I would not include unpublished articles

- Exclusion criteria: Not clear. Please explain.

- Outcome variable and data extraction: Please provide more details on the “predestined data extraction form”. The 1st section: How to assess the methodological quality of each study? The 2nd section: How to evaluate the comparability of the study group? The 3rd section: How to assess the outcome and statistical analysis?

Results

- What about incubation period and serial interval?

- please explain how to perform sensitivity analysis

Reviewer #2: Assessing the magnitude of asymptomatic and pre-symptomatic COVID-19 cases is important and currently relevant for COVID-19 pandemic over social and economic impacts.

Authors have here taken up a systematic review and meta-analysis on asymptomatic and pre-symptomatic and claimed symptom-based screening might fail to identify all potential SARS-CoV-2 infections. Finally, authors suggested to scale-up mass testing and targeting high risk populations to tackle the pandemic.

I anticipate that authors will be further address the suggested issue, mentioned below to improve the manuscript and its understanding to a broad readership.

1. Page-3: “Up to date, there are more than 22 million total cases with 780,000 deaths, worldwide (3).”, “Up to date” is not a right phrase to use here. Provide specific date of extracting data instead. Also, suggest to revise it as the study reporting the digits from long back in July, 2020.

2. Page-3: “Studies vary depending on the number of study participants recruited, the type of design employed and the country in which the study conducted.”, The sentence is not clear enough. What is the meaning of “the number of study participants recruited”? Need to revise the sentence for clarity. Further authors used the terms “Studies vary”, but not clearly mentioned in what context.

3. Page-3: Continuation of point 2 above, author need to present the research potential and motivation of the such review study in this paragraph before mentioning the objectives. Though author has mentioned the combining the exiting finding might strengthen the evidence.

4. Page-4: “In addition, we searched from the reference lists of the included studies to identify any other studies that may have been missed by our search strategy.”, I fell such statement is redundant as it’s a part of the review when reviewers need to assess the main text along with the title and abstract.

5. Page-4: “Our search was performed between the 1st of June and the 15th of July, 2020.”, the review time window is much backdated. Possibly suggest to update the time window to catch more data.

6. Page-4: “Inclusion criteria”, the term ‘Design’ is not really referring the design of the reviewed studies, instead it’s the outcome variable for which the review is subject to. It preferably better of mentioning “Estimates/ Estimates reported” instead of “Design”? Else clarify.

7. Page-4: “Exclusion criteria” need to mention clearly. Current form is not conveying what author wanted to mention. Need more detailed criteria.

8. Page-4: “Two experienced authors (MA and LY) extracted all essential data from the included studies using a predesigned data extraction form.”, hope these two review author did the search independently. If yes, please mention it else no need to mention specifically in the text. Independent search one if the tools to ensure the reproducibility of the results.

9. Page-5: “In addition, 10 articles were removed due to not extractable result and the outcome of interest not reported.”, This 10 studies were excluded after assessing the related “Text” of the articles. Need to be clearly mention it, as mentioned similarly for “titles and abstracts”.

10. Page-6: Table 1, “Study period” is not mentioned clearly for some of the studies. Need to clearly have a note for this in the table. Wonder, is it due to the missing information in the text.

11. Page-7: Author presented the magnitudes of ‘asymptomatic’ COVID-19 cases, wonder if it is included the ‘pre-symptomatic’ cases as well. Need clarification. Accordingly, the title of the manuscript should be revised.

12. Page-9: “We also compared the proportion of asymptomatic SARS-CoV-2 infections with previously emerged coronavirus pandemics.”, I wonder whether the MERS was a Pandemic? Otherwise, better to be specific by using ‘pandemics/epidemics’ or just simply ‘outbreaks”.

13. Page-9-10: “The result of this study suggest that contact and symptom based screening might fail to identify all potential SARS-CoV-2 infections.”, Author clamed this without any discussion in the main text on it. I would prefer to illustrate this in the discussion section first with the results and then suggest in the conclusion. Conclusion should be related to the results, otherwise take-home massage will be weaker to the readers.

14. Page-9-10: “Scale-up mass testing, which targeting high risk populations is recommended to tackle the pandemic. Continuation of the above points, ‘mass testing’ and ‘testing of the target high risk group’ are two different things. Prefer to revise or rearrange the sentence here.

15. Many of the references were in preprint now published should be updated.

16. The references are not according to the PloS One specification (“Vancouver” style). Need to be updated.

17. Finally, I would suggest to the authors to give some effort in the presentation and language of the manuscript to avoid the redundancy of text and improve the language for general readers.

6. PLOS authors have the option to publish the peer review history of their article (what does this mean?). If published, this will include your full peer review and any attached files.

Reviewer #1: No

Reviewer #2: No

---

## [Author Response · Author response to Decision Letter 0]

14 Dec 2020

Author's response to reviews

Title: Magnitude of asymptomatic COVID-19 cases throughout the course of infection: a systematic review and meta-analysis

Author’s email addresses 

MA: mulunehadis@gmail.com

LY: lielt.yismaw@gmail.com

MAA: agazhemoges@gmail.com

DBK: danibekele2009@gmail.com

BM: Belaynehmengist2008@gmail.com

BK: bekalukassiedmu@gmail.com

TYB: yemanu.tilahun@gmail.com

Date: 12 December 2020

Dear Editor,

We thank you for the chance to resubmit our revised version of the manuscript. Also, we would like to thank the reviewers for sharing their view and experience. The comments are very important that will improve the manuscript. The point-by-point responses for each of the comments are provided in the following pages. We hope that the revisions meet your standards and that the paper would be published in your journal. We look forward to working with you towards a final published product.

Sincerely,

Muluneh Alene, MPH

On behalf of co-authors

Point by point responses to queries 

Editor comments and suggestions 

This article aims to assess the magnitude of asymptomatic and pre-symptomatic COVID-19 cases which is essential and currently relevant for COVID-19 pandemic. Search terms may need to be refined to include as many as articles in the initial search. Will the authors consider to have pooled estimate stratified by age, setting such as hospital or community and study period to enrich the content. Also the exclusion criteria was not explicit mentioned. In terms of the language, authors are suggested to have substantial edit to arrive at the publishable quality.

Response

Thank you dear editor for your constructive comments and suggestions. In the revised version of the manuscript, we already refined the searching terms in order to include the existing potential studies. Unfortunately, the original studies were not report the magnitude of asymptomatic COVID-19 cases by age. Additionally, all studies included in this review are facility-based. We try to write the manuscript in acceptable quality of English language. 

Reviewer#1

Comments/suggestions #1

Severe Acute Respiratory Syndrome Corn babirusa 2, instead of Sever respiratory syndrome Coronavirus 2

Response#1

Thank you dear reviewer for your constructive suggestion and comments. The comments are very important to improve the manuscript. We used the standard WHO name of the novel coronavirus (Sever respiratory syndrome Coronavirus 2). 

Comments/suggestions #2

This systematic review and meta-analysis aims to determine the pooled magnitude of asymptomatic COVID-19 cases throughout the course of infection? What about total cases, pre-symptomatic cases? The course of infection or the course of outbreak?

Response#2

Thank you. The scope (objective) of this study was to determine the overall magnitude of asymptomatic COVID-19 cases throughout the course of infection using available evidences. In this review, we only consider patients who are asymptomatic (without sign and symptoms) in the course of infection, not outbreak.

Comments/suggestions #3

-Systematic review does not only include observational studies

- Study setting: worldwide? At hospital? At community?

- Population: All age group? Please provide age range.

- Publication status: I would not include unpublished articles

- Exclusion criteria: Not clear. Please explain.

- Outcome variable and data extraction: Please provide more details on the “predestined data extraction form”. The 1st section: How to assess the methodological quality of each study? The 2nd section: How to evaluate the comparability of the study group? The 3rd section: How to assess the outcome and statistical analysis?

Response#3

Thank you. We considered your comments and suggestions. We understand that systematic review and meta-analysis is not only for observational studies, but also it is applicable for experimental studies. However, in this study, we only consider observational articles. In addition, articles included in this review were facility-based. No restriction in age group. Furthermore, studies included in this systematic review were published and unpublished articles. Moreover, we revised the text in the section of exclusion criteria, outcome variable of the interest, and extraction form. Regarding, the quality assessment the following points are considered. 1) To assess the methodological quality of each study; we consider: representativeness of the sample, sample size, non-respondents, and ascertainment of the exposure (risk factor). 2) To evaluate the comparability of the study group; we consider: how studies control for the most important factor, and the study control for any additional factor.

3) To assess the outcome and statistical analysis; we consider: the statistical test used to analyze the data is clearly described and appropriate, and the measurement of the association is presented, including confidence intervals and the probability level.

Comments/suggestions #4

- What about incubation period and serial interval?

- Please explain how to perform sensitivity analysis

Response#4

Thank you. Incubation period is the time from infection occurred to the onset of signs and symptoms. Also, the serial interval is the time from illness onset in the primary case to illness onset in the secondary case. It also measured from pairs of cases with a clear infector–infectee relationship. In the revised version of the manuscript, we explained in detail how sensitivity analysis performed.

Reviewer#2

Comments/suggestions#1

Page-3: “Up to date, there are more than 22 million total cases with 780,000 deaths, worldwide.”, “Up to date” is not a right phrase to use here. Provide specific date of extracting data instead. Also, suggest to revise it as the study reporting the digits from long back in July, 2020.

Response #1

Dear reviewer, we thank you for your constructive comments and suggestions. The comments are very important to improve the manuscript. In the revised version of the manuscript, we provided the specific date with the current level COVID-19 cases and deaths. 

Comments/suggestions#2

Page-3: “Studies vary depending on the number of study participants recruited, the type of design employed and the country in which the study conducted.” The sentence is not clear enough. What is the meaning of “the number of study participants recruited”? Need to revise the sentence for clarity. Further authors used the terms “Studies vary”, but not clearly mentioned in what context.

Response#2

Thank you. Yes, we learnt that the sentence needs revision, and we already revised it in the revised version of the manuscript. In other word “the number of study participants recruited” means the sample size for a particular study. In addition, we describe in detail that in what context study varies.

Comments/suggestions#3

Page-3: Continuation of point 2 above, author need to present the research potential and motivation of such review study in this paragraph before mentioning the objectives. Though author has mentioned the combining the exiting finding might strengthen the evidence.

Response#3

Thank you. We try to explain further the potential and motivation of this study, in the revised form of the manuscript.

Comments/suggestions#4

Page-4: “In addition, we searched from the reference lists of the included studies to identify any other studies that may have been missed by our search strategy.”, I fell such statement is redundant as it’s a part of the review when reviewers need to assess the main text along with the title and abstract.

Response#4

Thank you. We considered to revise it.

Comments/suggestions#5

Page-4: “Our search was performed between the 1st of June and the 15th of July, 2020.” the review time window is much backdated. Possibly suggest to update the time window to catch more data.

Response#5

Thank you. We already updated the searching time.

Comments/suggestions#6

 Page-4: “Inclusion criteria”, the term ‘Design’ is not really referring the design of the reviewed studies, instead it’s the outcome variable for which the review is subject to. It preferably better of mentioning “Estimates/ Estimates reported” instead of “Design”? Else clarify.

Response#6

Based on the suggestion given, we mentioned “Estimates reported” instead of “Design”.

Comments/suggestions#7

Page-4: “Exclusion criteria” need to mention clearly. Current form is not conveying what author wanted to mention. Need more detailed criteria.

Response#7

In the revised version of the manuscript, we provided clearly the exclusion criteria of this study. 

Comments/suggestions#8

Page-4: “Two experienced authors (MA and LY) extracted all essential data from the included studies using a predesigned data extraction form.” hope these two review author did the search independently. If yes, please mention it else no need to mention specifically in the text. Independent search one if the tools to ensure the reproducibility of the results.

Response#8

Thank you, we removed the name of the authors upon your suggestion. 

Comments/suggestions#9

Page-5: “In addition, 10 articles were removed due to not extractable result and the outcome of interest not reported.” This 10 studies were excluded after assessing the related “Text” of the articles. Need to be clearly mention it, as mentioned similarly for “titles and abstracts”.

Response#9

In the revised manuscript, we explicitly mention the reason for excluding studies.

Comments/suggestions#10

Page-6: Table 1, “Study period” is not mentioned clearly for some of the studies. Need to clearly have a note for this in the table. Wonder, is it due to the missing information in the text.

Response#10

Thank you. Some original studies did not explicitly provided the study period (data collection period). 

Comments/suggestions#11

Page-7: Author presented the magnitudes of ‘asymptomatic’ COVID-19 cases, wonder if it is included the ‘pre-symptomatic’ cases as well. Need clarification. Accordingly, the title of the manuscript should be revised.

Response#11

Thank you. The outcome of interest (objective) of this study was to estimate the pooled magnitudes of asymptomatic COVID-19 cases throughout the course infection. We will come on that with another article.

Comments/suggestions#12

Page-9: “We also compared the proportion of asymptomatic SARS-CoV-2 infections with previously emerged coronavirus pandemics.” I wonder whether the MERS was a Pandemic. Otherwise, better to be specific by using ‘pandemics/epidemics’ or just simply ‘outbreaks”.

Response#12

Thank you. We restated it.

Comments/suggestions#13

Page-9-10: “The result of this study suggest that contact and symptom based screening might fail to identify all potential SARS-CoV-2 infections.” Author clamed this without any discussion in the main text on it. I would prefer to illustrate this in the discussion section first with the results and then suggest in the conclusion. Conclusion should be related to the results, otherwise take-home massage will be weaker to the readers.

Response#13

Thank you. We already revised it.

Comments/suggestions#14

Page-9-10: “Scale-up mass testing, which targeting high risk populations is recommended to tackle the pandemic. Continuation of the above points, ‘mass testing’ and ‘testing of the target high risk group’ are two different things. Prefer to revise or rearrange the sentence here.

Response#14

Thank you. We already revised it.

Comments/suggestions#15

Many of the references were in preprint now published should be updated.

16. The references are not according to the PloS One specification (“Vancouver” style). Need to be updated.

Response#15

Thank you. We already updated the references according to the PLOS ONE requirement.

Comments/suggestions#16

Finally, I would suggest to the authors to give some effort in the presentation and language of the manuscript to avoid the redundancy of text and improve the language for general readers.

Response#16

Thank you dear reviewer. We try to write the manuscript in acceptable quality of English language.

Thank you!!!

---

## [Decision Letter · Decision Letter 1]

22 Jan 2021

PONE-D-20-28251R1

Magnitude of asymptomatic COVID-19 cases throughout the course of infection: a systematic review and meta-analysis

PLOS ONE

Dear Dr. Addis,

Thank you for submitting your manuscript to PLOS ONE. After careful consideration, we feel that it has merit but does not fully meet PLOS ONE’s publication criteria as it currently stands. Therefore, we invite you to submit a revised version of the manuscript that addresses the points raised during the review process.

As suggested by the reviewer number 2, the authors are recommended to improve the language for general readers.

We look forward to receiving your revised manuscript.

Kind regards,

Kin On Kwok, Ph.D

Academic Editor

PLOS ONE

Additional Editor Comments (if provided):

Reviewers' comments:

Reviewer's Responses to Questions

**Comments to the Author**

1. If the authors have adequately addressed your comments raised in a previous round of review and you feel that this manuscript is now acceptable for publication, you may indicate that here to bypass the “Comments to the Author” section, enter your conflict of interest statement in the “Confidential to Editor” section, and submit your "Accept" recommendation.

Reviewer #1: All comments have been addressed

Reviewer #2: (No Response)

2. Is the manuscript technically sound, and do the data support the conclusions?

Reviewer #1: Yes

Reviewer #2: Partly

3. Has the statistical analysis been performed appropriately and rigorously? 

Reviewer #1: Yes

Reviewer #2: Yes

4. Have the authors made all data underlying the findings in their manuscript fully available?

Reviewer #1: Yes

Reviewer #2: Yes

5. Is the manuscript presented in an intelligible fashion and written in standard English?

Reviewer #1: Yes

Reviewer #2: No

6. Review Comments to the Author

Reviewer #1: (No Response)

Reviewer #2: Authors tried to revise the manuscript but unable to improve the manuscript adequately. I further suggest authors to make the text precise and continuous for readers’ interest. In fact, authors missed/ avoided some of the comments/suggestions to address, here I strongly suggest to take up these seriously and revise the text extensively. I added those in further comments below as “Comments/Suggestions on Response #xx in R1”.

###################

Reviewer#2

Comments/suggestions#1: Page-3: “Up to date, there are more than 22 million total cases with 780,000 deaths, worldwide.”, “Up to date” is not a right phrase to use here. Provide specific date of extracting data instead. Also, suggest to revise it as the study reporting the digits from long back in July, 2020.

Response #1: Dear reviewer, we thank you for your constructive comments and suggestions. The comments are very important to improve the manuscript. In the revised version of the manuscript, we provided the specific date with the current level COVID-19 cases and deaths.

Comments/Suggestions on Response #1 in R1: Response and revision acceptable.

Comments/suggestions#2: Page-3: “Studies vary depending on the number of study participants recruited, the type of design employed and the country in which the study conducted.” The sentence is not clear enough. What is the meaning of “the number of study participants recruited”? Need to revise the sentence for clarity. Further authors used the terms “Studies vary”, but not clearly mentioned in what context.

Response#2: Thank you. Yes, we learnt that the sentence needs revision, and we already revised it in the revised version of the manuscript. In other word “the number of study participants recruited” means the sample size for a particular study. In addition, we describe in detail that in what context study varies.

Comments/Suggestions on Response #2 in R1: Still text needs to be revised. Please simplify as “Studies vary depending on the number of participants recruited, the type of design employed and the country in which the study conducted.”

Comments/suggestions#3: Page-3: Continuation of point 2 above, author need to present the research potential and motivation of such review study in this paragraph before mentioning the objectives. Though author has mentioned the combining the exiting finding might strengthen the evidence. Response#3 Thank you. We try to explain further the potential and motivation of this study, in the revised form of the manuscript.

Comments/Suggestions on Response #3 in R1: I was expecting more on motivations here. Specially to indicate why authors took up this study.

Comments/suggestions#4: Page-4: “In addition, we searched from the reference lists of the included studies to identify any other studies that may have been missed by our search strategy.”, I fell such statement is redundant as it’s a part of the review when reviewers need to assess the main text along with the title and abstract.

Response#4: Thank you. We considered to revise it.

Comments/Suggestions on Response #4 in R1: What are the revision here? It was just a suggestion. If you want to keep the sentence, no problem, still make sense. But could not understand what revision has been made in this context as claimed by author in revised version?

Comments/suggestions#5: Page-4: “Our search was performed between the 1st of June and the 15th of July, 2020.” the review time window is much backdated. Possibly suggest to update the time window to catch more data.

Response#5: Thank you. We already updated the searching time.

Comments/Suggestions on Response #5 in R1: Response and revision acceptable.

Comments/suggestions#6: Page-4: “Inclusion criteria”, the term ‘Design’ is not really referring the design of the reviewed studies, instead it’s the outcome variable for which the review is subject to. It preferably better of mentioning “Estimates/ Estimates reported” instead of “Design”? Else clarify.

Response#6: Based on the suggestion given, we mentioned “Estimates reported” instead of “Design”.

Comments/Suggestions on Response #6 in R1: Response and revision acceptable.

Comments/suggestions#7: Page-4: “Exclusion criteria” need to mention clearly. Current form is not conveying what author wanted to mention. Need more detailed criteria.

Response#7: In the revised version of the manuscript, we provided clearly the exclusion criteria of this study.

Comments/Suggestions on Response #7 in R1: Still not clear. Authors should not use the subjective term ‘fully’ here. What did the authors mean by ‘fully’?

Comments/suggestions#8: Page-4: “Two experienced authors (MA and LY) extracted all essential data from the included studies using a predesigned data extraction form.” hope these two review author did the search independently. If yes, please mention it else no need to mention specifically in the text. Independent search one if the tools to ensure the reproducibility of the results.

Response#8: Thank you, we removed the name of the authors upon your suggestion.

Comments/Suggestions on Response #8 in R1: Not sure, whether I was able to convey the issue to the authors by the comment? Removing the authors’ names were not the suggestion. Here I wanted to know whether two authors had performed the data extraction from the studies included independently or not? Generally, in systematic review and metanalysis the two (or more) authors performed such data extraction independently to ensure the reproducibility. Further, how did the authors settled the mismatches? should be clearly mentioned in the text.

Comments/suggestions#9: Page-5: “In addition, 10 articles were removed due to not extractable result and the outcome of interest not reported.” This 10 studies were excluded after assessing the related “Text” of the articles. Need to be clearly mention it, as mentioned similarly for “titles and abstracts”. Response#9: In the revised manuscript, we explicitly mention the reason for excluding studies.

Comments/Suggestions on Response #6 in R1: Response and revision acceptable.

Comments/suggestions#10: Page-6: Table 1, “Study period” is not mentioned clearly for some of the studies. Need to clearly have a note for this in the table. Wonder, is it due to the missing information in the text.

Response#10: Thank you. Some original studies did not explicitly provided the study period (data collection period).

Comments/Suggestions on Response #10 in R1: Please mention these as a footnote. Indicate with notation ‘*’ in the table text and explain it in the table footnote.

Comments/suggestions#11: Page-7: Author presented the magnitudes of ‘asymptomatic’ COVID-19 cases, wonder if it is included the ‘pre-symptomatic’ cases as well. Need clarification. Accordingly, the title of the manuscript should be revised.

Response#11: Thank you. The outcome of interest (objective) of this study was to estimate the pooled magnitudes of asymptomatic COVID-19 cases throughout the course infection. We will come on that with another article.

Comments/Suggestions on Response #11 in R1: Response and revision acceptable.

Comments/suggestions#12: Page-9: “We also compared the proportion of asymptomatic SARS-CoV-2 infections with previously emerged coronavirus pandemics.” I wonder whether the MERS was a Pandemic. Otherwise, better to be specific by using ‘pandemics/epidemics’ or just simply ‘outbreaks”.

Response#12: Thank you. We restated it.

Comments/Suggestions on Response #12 in R1: Response and revision acceptable.

Comments/suggestions#13 Page-9-10: “The result of this study suggest that contact and symptom based screening might fail to identify all potential SARS-CoV-2 infections.” Author clamed this without any discussion in the main text on it. I would prefer to illustrate this in the discussion section first with the results and then suggest in the conclusion. Conclusion should be related to the results, otherwise take-home massage will be weaker to the readers.

Response#13: Thank you. We already revised it.

Comments/Suggestions on Response #13 in R1: Actually authors has removed the sentence. Response and revision acceptable.

Comments/suggestions#14: Page-9-10: “Scale-up mass testing, which targeting high risk populations is recommended to tackle the pandemic. Continuation of the above points, ‘mass testing’ and ‘testing of the target high risk group’ are two different things. Prefer to revise or rearrange the sentence here. Response#14: Thank you. We already revised it.

Comments/Suggestions on Response #14 in R1: Response and revision acceptable.

Comments/suggestions#15: Many of the references were in preprint now published should be updated. 16. The references are not according to the PloS One specification (“Vancouver” style). Need to be updated.

Response#15: Thank you. We already updated the references according to the PLOS ONE requirement.

Comments/Suggestions on Response #15 in R1: Response and revision acceptable.

Comments/suggestions#16: Finally, I would suggest to the authors to give some effort in the presentation and language of the manuscript to avoid the redundancy of text and improve the language for general readers.

Response#16: Thank you dear reviewer. We try to write the manuscript in acceptable quality of English language.

Comments/Suggestions on Response #16 in R1: Still the text need several improvement in English language, specially in terms of preciseness.

7. PLOS authors have the option to publish the peer review history of their article (what does this mean?). If published, this will include your full peer review and any attached files.

Reviewer #1: No

Reviewer #2: **Yes: **Sheikh Taslim Ali

---

## [Author Response · Author response to Decision Letter 1]

20 Feb 2021

Author's response to reviews

Title: Magnitude of asymptomatic COVID-19 cases throughout the course of infection: a systematic review and meta-analysis

Author’s email addresses 

MA: mulunehadis@gmail.com

LY: lielt.yismaw@gmail.com

MAA: agazhemoges@gmail.com

DBK: danibekele2009@gmail.com

BM: Belaynehmengist2008@gmail.com

BK: bekalukassiedmu@gmail.com

TYB: yemanu.tilahun@gmail.com

Date: 20 February 2021

Dear Editor,

We thank you for the chance to resubmit our revised version of the manuscript. Also, we would like to thank the reviewers for sharing their view and experience. The comments are very important that will improve the manuscript. The point-by-point responses for each of the comments are provided in the following pages. We hope that the revisions meet your standards and that the paper would be published in your journal. We look forward to working with you towards a final published product.

Sincerely,

Muluneh Alene, MPH

On behalf of co-authors

Point by point responses to queries 

Reviewer#2

Comments/suggestions #1

Comments/Suggestions on Response #2 in R1: Still text needs to be revised. Please simplify as “Studies vary depending on the number of participants recruited, the type of design employed and the country in which the study conducted.”

Response#1

Dear reviewer thank you so much for your constructive comments. We have revised it.

Comments/suggestions #2

Comments/Suggestions on Response #3 in R1: I was expecting more on motivations here. Specially to indicate why authors took up this study.

Response#2

Thank you. In the revised manuscript, we added more on the motivation of this study. 

Comments/suggestions #3

Comments/Suggestions on Response #4 in R1: What are the revision here? It was just a suggestion. If you want to keep the sentence, no problem, still make sense. But could not understand what revision has been made in this context as claimed by author in revised version?

Response#3

Thank you. We keep the sentence as it is.

Comments/suggestions #4

Comments/Suggestions on Response #7 in R1: Still not clear. Authors should not use the subjective term ‘fully’ here. What did the authors mean by ‘fully’?

Response#4

Thanks. We already addressed it.

Comments/suggestions #5

Comments/Suggestions on Response #8 in R1: Not sure, whether I was able to convey the issue to the authors by the comment? Removing the authors’ names were not the suggestion. Here I wanted to know whether two authors had performed the data extraction from the studies included independently or not. Generally, in systematic review and met analysis the two (or more) authors performed such data extraction independently to ensure the reproducibility. Further, how did the authors settled the mismatches? Should be clearly mentioned in the text.

Response#5

Thanks. We have addressed it.

Comments/suggestions #6

Comments/Suggestions on Response #10 in R1: Please mention these as a footnote. Indicate with notation ‘*’ in the table text and explain it in the table footnote.

Response#6

Thanks. We considered your suggestion.

Comments/suggestions #7

Comments/Suggestions on Response #16 in R1: Still the text need several improvement in English language, especially in terms of preciseness.

Response#7

Thank you. In the revised manuscript, we have improved the language of the manuscript using English language professionals.

Thank you!!!

---

## [Editor Report · Decision Letter 2]

11 Mar 2021

Magnitude of asymptomatic COVID-19 cases throughout the course of infection: a systematic review and meta-analysis

PONE-D-20-28251R2

Dear Dr. Addis,

We’re pleased to inform you that your manuscript has been judged scientifically suitable for publication and will be formally accepted for publication once it meets all outstanding technical requirements.

Kind regards,

Kin On Kwok, Ph.D

Academic Editor

PLOS ONE

---

## [Editor Report · Acceptance letter]

15 Mar 2021

PONE-D-20-28251R2 

Magnitude of asymptomatic COVID-19 cases throughout the course of infection: a systematic review and meta-analysis 

Dear Dr. Alene:

I'm pleased to inform you that your manuscript has been deemed suitable for publication in PLOS ONE. Congratulations! Your manuscript is now with our production department. 

Kind regards, 

on behalf of

Dr. Kin On Kwok 

Academic Editor

PLOS ONE